# Generational Differences in Food Choices and Consumer Behaviors in the Context of Sustainable Development

**DOI:** 10.3390/foods13040521

**Published:** 2024-02-08

**Authors:** Marta Makowska, Rafał Boguszewski, Arleta Hrehorowicz

**Affiliations:** 1Department of Economic Psychology, Kozminski University, Jagiellońska 57/59 St., 03-301 Warsaw, Poland; mmakowska@kozminski.edu.pl; 2Institute of Sociological Sciences and Pedagogy, Warsaw University of Life Sciences, Nowoursynowska 166 St., 02-787 Warsaw, Poland; arleta_hrehorowicz@sggw.edu.pl

**Keywords:** Poland, generations, sustainability

## Abstract

(1) Background: This study examines diverse food choices and sustainable consumption behavior across four generations of Poles: Baby Boomers, Generation X, Generation Y, and Generation Z. (2) Methods: The research employed an online survey with a quota sample of 1000 individuals, representing the Polish population in terms of gender, residence, and education (categorized into two groups). (3) Results: For Poles, the crucial factors in food selection were product quality (69.0%), price (68.6%), and information about the product’s composition (41.0%). Older cohorts (Baby Boomers and Generation X) prioritized food quality, while younger generations (Generation Y and Generation Z) emphasized price. Statistically significant differences emerged between generations regarding the product’s country of origin, with older generations emphasizing its importance more. The oldest group (Baby Boomers) considered ecological certification most important. As much as 85.9% of Poles checked product expiration dates while buying products, and 82.8% bought only as much as they needed. Baby Boomers most often exhibited these two behaviors and can be considered the most responsible among all generations. (4) Conclusions: This article highlights the importance of comprehensive sustainability efforts in the context of food choices. It takes into account the socio-cultural and historical circumstances of each generation that influence their consumer habits.

## 1. Introduction

Food is a fundamental human need [1] and an essential product for daily consumption [2]. Food choices have a significant impact on the environment [3,4]. It is important to make responsible decisions regarding food production, food packaging, and food waste, as the daily choices individuals make about what to eat, where to eat, how to prepare food, how much to eat, and where to buy food ultimately affect our environment [3].

In connection with population growth, the demand for food worldwide is rapidly increasing, leading to agricultural expansion [5] which, in turn, results in environmental degradation on multiple levels. This is related to the excessive use of chemicals, which has a destructive impact on both human health and ecosystems [5]. It contributes to the consumption of vast amounts of energy and natural resources, as well as the generation of significant pollution. One-third of worldwide anthropogenic greenhouse gas emissions can be attributed to food systems [6]. Recognizing the significance of food in the context of sustainable development is crucial for building appropriate practices aimed at addressing global challenges. Sustainable development involves developing various strategies that aim to promote environmental conservation and social equity. In the case of food, this will include promoting local food markets [7,8] reducing food waste through proper storage, conscious purchasing, well-planned distribution, and adopting more sustainable packaging [3,9,10]. It is also important to promote a plant-based diet, as animal-based products and meat are the foods with the most significant environmental burden [4]. Therefore, sustainable consumption is a very crucial element in taking care of the well-being of the environment.

The symposium in Oslo, which took place in 1994, provided a definition of sustainable consumption as the practice of using goods and services that meet basic needs while simultaneously improving the quality of life. This is achieved by minimizing the use of natural resources, reducing toxic materials, and limiting waste and pollutant emissions throughout the life cycle in a way that does not pose a threat to the needs of future generations. Sustainable consumption is associated with several key issues, such as meeting needs, improving the quality of life, the efficient use of resources, expanding the use of renewable energy sources, reducing the amount of generated waste, and considering the life cycle perspective and capital aspect [11]. According to the European Environment Agency, the main goals and challenges associated with sustainable consumption involve improving products and modifying consumption patterns [12]. Dermody et al. [13] believe that an orientation towards sustainable consumption involves the ability to: (a) equitably share resources, thereby improving people’s quality of life; (b) consider the needs of future generations by limiting excessive consumption; and (c) reduce the harmful impact of excessive consumption on the environment.

Transitioning to more sustainable food systems is a key element of the European Union’s strategy, as evidenced by the debate on the European Green Deal [14]. One of the goals of this strategy is to ensure that the EU’s food system becomes a global standard for sustainability, benefiting both the environment and society. In the context of social equality, sustainable food systems can contribute to greater fairness in the food supply chain, increase farmers’ incomes, and improve the quality of food available to consumers [15]. Including environmental issues in the assessment of import tolerance applications, as well as EU cooperation with trading partners, particularly developing countries, towards safer plant protection methods, can help evenly distribute the benefits of sustainable food systems. The EU is also involved in promoting a global transformation towards sustainable food systems through international standard-setting bodies and international events [16].

Education is another key element of the EU’s strategy. Social equality and social justice issues should be incorporated into educational programs related to sustainable food systems. Educational content should consider diverse social and economic perspectives, contributing to a better understanding and more effective implementation of sustainable practices in food systems [17]. Education plays a crucial role in fostering confidence and ensuring safety in daily food consumption by providing people with information about emerging food threats and effective actions at the household level [18]. In 2006, The World Health Organization developed guidelines for safer food, aimed not only at drawing attention to potential food-related hazards but also at making consumers aware of the necessity to adhere to strict hygiene standards during meal preparation [19].

In the context of research on sustainable consumption, a trend called the minimalist lifestyle, also known as voluntary simplicity or a low-consumption lifestyle, can be found. This is an approach assuming that consumers consciously limit their material consumption, seeking to free up financial resources and time while simultaneously finding satisfaction in non-material aspects of life [20,21]. There are studies that link minimalism with well-being [20,22], as well as with greater life satisfaction [23]. In the realm of food, a form of such minimalism involves adopting various diets such as veganism, vegetarianism, embracing certain lifestyles like freeganism, and opting for slow food.

Consumer values, beliefs about food, and behaviors related to its consumption are culturally determined [24]. Deeply ingrained food choices and habits, often varying across generations, impact the environment, the economy, public health, and the quality of life for future generations; therefore, research on food choices and sustainable consumption behaviors in the context of different generations is crucial to understand the extent to which various age groups contribute to generating threats related to the sustainable future of our planet. Our study will contribute to a better understanding of this issue by presenting the attitudes of different generations towards sustainable food consumption in Poland. The main goal of the article is to describe the differences in food choices and sustainable consumption behavior among the four generations. To the best of our knowledge, this is the first study examining this relationship in Polish society, which has its specificity due to its specific historical and political conditions as a post-communist country. Social and cultural conditions are significant variables for consumer behavior [25]. This study aimed to address the following main research questions: (a) Do generations differ in their motivations for food purchases, and if so, how? (b) Which generations are closest to sustainable consumption in the context of everyday consumer behaviors?

## 2. Theoretical Framework

### 2.1. Generational Differences in Food Buying Motives

A generational cohort is a group of individuals who share similar experiences and exhibit unique characteristics associated with those experiences. Shared experiences can also influence similar consumption-related behaviors [25]. It must not be forgotten that technology and media also influence the food choices and customer behaviors of different generations. Primarily, younger generations are more inclined to embrace modern technological solutions [26]. This can also manifest in using various applications that prevent food waste through meal planning or buying food that would otherwise be discarded (e.g., Too Good To Go).

The time intervals used to distinguish different generations are debatable. Sociologists have used various approaches—such as Mannheim in the 1950s, who employed thirty-year intervals [27], and Strauss and Howe, who used fifteen-year and thirty-three-year intervals [28]. Generational divisions are arbitrary, and the boundaries between them are unclear. For the purposes of this article, we will focus on the distinction used by Twenge [29]: (a) Baby Boomers (born between 1946 and 1964), (b) Generation X (born between 1965 and 1979), (c) Generation Y (born between 1980 and 1994), and (d) Generation Z (born after 1994). Baby Boomers, or individuals born just after World War II, grew up during a period when Poland was rebuilding from wartime destruction. Their youth was spent under the shadow of communism, which influenced their pragmatic approach to life. Stability and security were fundamental values for them. In the context of shopping, Baby Boomers often look for products that are durable and functional, remembering times when access to goods was limited [30].

Generation X refers to individuals who experienced the effects of the economic crisis in the 1970s in Poland. Their worldview was also shaped by the communist political system at the time. As Bielińska-Dusza rightly points out, as this generation reached maturity in the 1990s (a time of socio-economic transformation), ‘it is not typical of its counterparts elsewhere and has few features in common with the prototype from the United States’ [31]. Observing both social and economic changes made them adapt to the evolving reality, yet they remained somewhat skeptical towards excessive consumption. Although they recognized the emerging new opportunities associated, for instance, with the increased availability of many products and services, they did not forget the period of limitations and scarcity [32]. 

The Generation Y, also known as millennials, are considered thoughtful consumers, who are engaged in ethical consumerism [33]. They are less influenced by brand-directed marketing and often rely on recommendations and reviews found on the Internet, social media, or from friends [34]. McGlone et al. indicate that millennials see the need to make the world a better place to live [35]. Around half of the millennials are willing to pay more for high-quality products that will last for many years [36]. Other key factor is the convenience of shopping, with Generation Y being the most active users of online marketplaces [34].

Generation Z, as Agrawal [37] claims, was defined by the COVID-19 pandemic crisis, characterized by inflation, a higher cost of living, lower incomes, the burden of education loans, and disruptions in supply chains. This situation has led to emotional and social distress, influencing their shopping habits as well. Generation Z is more inclined to seek authenticity and transparency from brands, which means they prefer companies that are transparent, ethical, and eco-friendly [38]. They are true digital natives, and social media has a significant impact on their purchasing decisions, as do the opinions and recommendations of influencers [37]. 

Shepherd and Raats [39] indicate that people may have various motivations related to food choices. They divide these motivations into the following: (a) concerns about the food itself (ingredients and properties) with the aim of satisfying human needs (hunger and thirst), (b) those related to the individual, including perception (aroma and taste) and psychological factors (mood, emotions, and beliefs), and (c) the involvement of external economic and social conditions (price and social influences). In the psychological literature on customer behavior, many factors are mentioned that influence motivations related to product choices [40,41]. The issue of food choices is complex and requires an interdisciplinary approach [42]. Among its determinants, researchers from various disciplines mention product quality [24,43], price [39,44,45,46], labeling information [39,47], promotions [45,48], habit [42], country of origin [49], the presence of ecological certification for the product [50,51], opinions of family/friends [52,53], product appearance [54,55], and product advertising [46].

Recent events in Poland and neighboring countries have had a significant impact on Poles’ sustainable consumption behaviors. The COVID-19 pandemic has highlighted the vulnerability of food security and nutritional progress [56], while the ongoing war in Ukraine has changed the supply chains of agricultural products from both Ukraine and Russia [57]. The sense of food security among Poles undoubtedly transformed under the influence of these events. Additionally, the high inflation could be affecting their food-related decisions. In February 2023, inflation in Poland was the highest since December 1996, reaching 18.4%. The inflation rate for food was even higher at 24% [48]. The CBOS study shows that inflation has led Poles to make purchases in advance more frequently [58]. This is an undesirable phenomenon from the perspective of sustainable consumption [59]. Inflation can influence the quantity and quality of products purchased. It can lead to changes in consumer spending habits, with individuals opting for more affordable food options or adjusting their consumption patterns [44]. Inflation can also have different influences on customer behaviors of different generations, as their financial situation is different. In the worst financial situation are the youngest adult people whose group unemployment rate is the highest, and they have needs such as buying a flat or house to start a family.

Since May 2004, Poland has been a part of the European Union. The younger generations feel more like citizens of Europe [60] or even the world [61] than older individuals. They are not as patriotic as older Poles [61].

Based on our knowledge of the historical conditions in which the different generations grew up, as well as on the literature and data from other studies that deal with the consumer and socio-political attitudes and behavior of Poles, we decided to verify the following research hypotheses.

**Hypothesis** **1.**
*The price of the food product will be more important for the youngest generation, Generation Z, than for the older generations (Baby boomers, X, and Y).*


**Hypothesis** **2.**
*The country of origin will be a more important motive for buying food for Baby Boomers than for other younger generations (X, Y, and Z).*


### 2.2. Generational Differences in Declarations of Sustainable Food Customer Behavior

Poland, as a post-communist country, has a unique history and culture that have profoundly influenced its society and economy. After World War II, Poland fell under Soviet influence, leading to the establishment of a communist government. Elections during this period were controlled by the USSR, serving as a tool to legitimize Soviet hegemony over Poland. Communist rule lasted until the late 1980s when the ‘Solidarity’ movement emerged, contributing to political and economic transformations in the country [62].

In the 1990s, Poland underwent a transition from a planned to a market economy, part of a broader process of reforms in post-communist countries. This period was marked by a ‘shock therapy’ program aimed at rapidly introducing free-market reforms. Though it ultimately led to economic growth, the initial years of transformation were challenging for many Poles, including lower living standards and a significant increase in social inequality [63,64].

From a generational perspective, these historical experiences significantly impact purchasing decisions. Older generations (Baby Boomers and Generation X) who remember communist times may have mixed feelings about consumption and material values. They might value the stability and security of the communist era, yet appreciate the greater choice and product availability resulting from the transition to a market economy [65].

The oldest generation may be linked to traditional consumption habits and less exposure to contemporary discussions on sustainability. With age, it becomes increasingly difficult to change one’s habits. For this reason, we formulated the following hypothesis:

**Hypothesis** **3.**
*The Baby Boomer generation less frequently declares sustainable consumer behaviors than younger generations (X, Y, and Z).*


Generation Z has received a fairly good education regarding sustainability, and awareness of climate change is instilled in them. They also exhibit a willingness to take action for the environment and sustainable development. For this reason, we formulated the following hypothesis:

**Hypothesis** **4.**
*Generation Z more frequently declares all sustainable consumer behaviors compared to older generations (Y, X, and Baby Boomer).*


We have considered the following activities as indicators of sustainable consumer behavior:(1)Before purchasing, I check the expiration date of the product;(2)I buy thoughtfully and only as much as I need at the moment;(3)I am willing to pay 10% more and choose a product labeled as eco-friendly than a cheaper product without such labeling;(4)Before purchasing, I check whether the product or its packaging is biodegradable (subject to recycling).

## 3. Materials and Methods

This sociological study was conducted using an online survey in February 2023 on the SW Research internet panel. The questionnaire was a proprietary tool created for this study by the authors and consisted of 11 questions, including 5 related to respondents’ socio-demographic characteristics. Throughout the development of the questionnaire we followed the procedures outlined by Malhotra (1999). The validation of the questionnaire consisted of three stages: (1) content validation (the content of the questionnaire was reviewed by experts for the purposes of the correct operationalization of the questions); (2) a preliminary study was conducted among 10 individuals, and their feedback was taken into account in the final version of the questionnaire; and (3) a reliability analysis of the indicators measured by Cronbach’s alpha test, which showed an acceptable test value for the indicators measuring sustainable customer behaviors (0.6).

It took participants approximately 5 min to complete the survey. In the end, 137 responses were excluded from the analysis for reasons such as: (a) too short of a completion time; (b) questionnaires were filled out systematically (e.g., the same option chosen throughout the response set); (c) a lack of logic in demographic questions; and (d) systemic data gaps. The final analysis was conducted on a sample of 1000 individuals. The questionnaire and the study dataset are available upon request to the corresponding author. The sample reflected the demographic structure of Polish citizens in terms of gender, education (2 groups), and province. Each identified generation comprised 250 individuals, allowing for more precise comparisons between groups, with the possibility of controlling for variables other than age. Ex ante clustering also simplified comparative analyses and strengthened their statistical power [66].

The Ethics Committee was not included as it was not obligatory for online surveys in Poland at the time of conducting our research. All participants provided informed consent to participate in the study and could withdraw at any time if any question seemed too sensitive. The study was conducted by one of the largest forums in Poland, where participants are registered users, and for participating in surveys, they receive points that they can later exchange for selected rewards from the reward pool. This forum meets the quality standards of the PKJPA (Polish Quality Certificate in Market and Opinion Research) and is a member of the PTBRIO (Polish Society of Market and Opinion Researchers) and ESOMAR (European Society for Opinion and Marketing Research).

A statistical analysis of the results was performed using SPSS software version 29. When presenting the results, we used frequencies as well as measures of central tendency and dispersion. To examine statistical relationships between generations regarding differences in food purchase motivations (the verification of Hypotheses 1 and 2), the Chi^2^ test was employed. Subsequently, in cases where a statistically significant relationship was obtained, the observed and expected values were checked, and standard residuals were calculated.

For analyses related to sustainable consumption (the verification of Hypotheses 3 and 4), we conducted a comparison of means using one-way ANOVA. Subsequently, in cases where ANOVA indicated a statistically significant difference between the means in individual generations, post hoc tests were conducted to determine which generations differed from each other. Accordingly, in cases where the assumption of variance equality was met, the Tukey post hoc test was applied, and where this assumption was not met, the Games–Howell post hoc test was used.

## 4. Results

According to declarations, 69% of Poles prioritize the quality of products when selecting food. The price of the product was the motive for buying was indicated by 68.6% respondents. Product quality is the most important feature for Generations X and Baby Boomers, while for Generations Z and Y, the most important feature is the price. In the case of both motivations, the Chi^2^ test showed a statistically significant difference.

Information about the product’s composition on the packaging is important for 41% of those surveyed. Additionally, in this regard, the differences between generations proved to be statistically significant. The Baby Boomer generation most often takes this feature into account, while Generations Y and X do so the least.

Promotions as a motivation for purchasing food products was indicated by a total of 30.7% of respondents, while habit as motive was considered by 24.3%. In both cases, there were no statistically significant differences between the generational cohorts.

The motivating factor that statistically differentiated customer behaviors of different age groups turned out to be the country of origin of the products. Overall, 17.7% of respondents indicated this feature. The standard residuals show that in this case, Generation Z chooses it statistically significantly less than expected (sr = −2.1), while the oldest (Baby Boomers) statistically significantly more than expected (sr = 2.2).

The presence of an ecological certificate is important for 9.7% of respondents. In this regard, there are also statistically significant differences between generations. This motivation is most often considered by individuals from the Baby Boomer generation, who chose it statistically significantly more frequently than expected (sr = 2.4).

Other features, such as the opinion of family and friends, the appearance of the product, its advertising, and others, were less frequently mentioned than those listed above and did not significantly differentiate the generations. The details are presented in Table 1.

The declarations of sustainable consumption behaviors by Poles were measured on a five-point Likert scale. For the purposes of this article, only statements (see Table 2) related to sustainable food consumption were selected from a larger set of statements. A significant majority of Poles, 85.9%, declare that they check the expiration date of a product. The oldest generation (Baby Boomers) declared this behavior most often. Post hoc analyses presented in Table 3 indicated that there is a statistically significant difference in declarations of this behavior between the Baby Boomer generation and Generation X, as well as between the Baby Boomer generation and Generation Z.

As much as 82.8% of Poles declare that they buy considerately, only as much as they need at the moment; therefore, few people buy things in excess. This behavior is most often exhibited by the oldest group (Baby Boomers), and additional analyses (see Table 3) showed that there is a statistically significant difference between them and the youngest generations, Generations Y and Z.

Other declarations of behaviors—willingness to pay 10% more for an ecological product and checking whether a product or its packaging is biodegradable—did not show any differentiation based on the generational affiliation of the respondents; however, in both cases, it can be observed that the number of declarations of definitely yes and rather yes increases with the age of the respondents.

Based on all four declarations of sustainable consumer behaviors, we have created a composite index measured on a scale from 1 to 5, where higher values indicate more sustainable consumption. It turns out that the average value of the index increases with the age of the respondents and is highest in the Baby Boomer generation, while being lowest among representatives of Generation Z (see Table 4).

The post hoc analyses, as shown in Table 5, indicated that there is a statistically significant difference in behaviors indicative of sustainable consumption between the Baby Boomer generation and Generation Y, as well as between the Baby Boomer generation and Generation Z.

## 5. Discussion

According to the Global Food Security Index 2022 report, Poland ranked 15th out of the 113 surveyed countries in terms of food quality and safety. The same index indicated that since 2015, the price of food in Poland has increased on average more than in other countries [67]. These are important pieces of information, as the quality of food products turned out to be the most significant motivation for buying in our study (69.0%), with the next criterion being the price (68.6%). The same order of dominant motivations for buying food was obtained in a study conducted in the neighboring country of the Czech Republic [68].

Perceived quality can be defined as ‘the degree to which a product fulfills its functions, given the needs of consumers’ [43]. When evaluating the quality of products, consumers consider a range of attributes in the food [24], and different consumers may assign varying importance to individual attributes [43]. Therefore, it is difficult to explain precisely what quality is; however, we know that it is an important criterion when purchasing food [24]. Significant emphasis on price may be associated with inflation in Poland and substantial increases in food prices [69] related to the COVID-19 pandemic, followed by the conflict in Ukraine. Our study revealed differences between generations. The Baby Boomer generation and Generation X are primarily driven by quality, while Generations Y and Z prioritize price. The approach of younger generations in this aspect may seem concerning, especially since research indicates that a healthy diet is more expensive than an unhealthy one [70,71]. This could be attributed to the fact that Generations Y and Z have a less stable financial situation compared to Generation Baby Boomers, who have already raised their children, paid off housing loans, and consequently have fewer financial burdens [72]. They are also more brand loyal and willing to pay more if they consider a product to be of higher quality, as explained by Wong [49].

Information about the composition of a product is a significant motivator for 41% of the surveyed individuals. In Poland, the key document regarding food labeling is the European Parliament and Council Regulation (EU) No 1169/2011 [73]. Information about the ingredients of a product is most crucial for the oldest generation (Baby Boomers). The ingredients can be important for individuals with specific health conditions or on particular diets. Since health tends to decline with age, this phenomenon may be an attempt to explain it, though it requires further research. Similarly, the connection between quality (most important for the Baby Boomer generation) and the ingredients of a product calls for future reflection.

In addition to these three main motivators for purchasing food, statistically significant differences between generations have also emerged when choosing reasons such as the country of origin of the product and the presence of an ecological certificate.

Producers in Poland can use the ‘Produkt Polski’ (‘Polish Product’) label only when less than 25% of the total weight of all ingredients did not originate from within the country [74]. The country of origin was identified as one of the three main motivators for purchasing food, as indicated by 17.7% of the respondents. The oldest generation, Baby Boomers [75], most frequently choose the country of origin as a motivator for purchasing food. This can be additionally explained by patriotism, as the oldest generation (Baby Boomers) lived through the socialist era, during which it was challenging to purchase foreign goods, and there were restrictions on traveling abroad. Furthermore, they belong to a generation that is more conservative in their views and less inclined towards change. A similar explanation was provided in a study on another post-communist country [49].

On labels, producers can also include symbols and certifications indicating distinctive features of the product. In Poland, for example, the ‘Poznaj dobrą żywność’ (‘Get to know good food’) symbol is used. Decisions about issuing the certification are made by the relevant minister for a period of three years [74]. The presence of an ecological certificate was chosen by 9.7% of the respondents, with individuals from the Baby Boomer generation once again doing so most frequently. Perhaps this is because we consider the certificate as a credibility-enhancing element; we believe that certifications and quality marks signify high-quality products, and the producer operates in accordance with norms and standards. The relation between the presence of a certificate and quality should also be a subject of future research.

Consuming products before their expiration date is one of the responsible consumer behaviors. It is important to learn to distinguish between ‘best by’ and ‘use by’ labels [59]. The placement of dates on products by manufacturers can lead to food waste because people may lack the skills to properly understand labeling and its impact on the quality and safety of food [76,77]. Research is being conducted on how food products should be labeled with dates to minimize food waste [35]. In Poland, it is slightly easier because only a label indicating ‘use by’ is employed [78]. A significant number of Poles (85.9%) stated that they check the expiration date before purchasing a product. One could presume that they do so out of concern for their health (food safety) or for the product’s quality. This assumption is supported by our finding that the Baby Boomer generation most frequently checks the expiration date. They are also the generation facing the most significant health issues, and as we demonstrated earlier, they are oriented towards quality. However, this issue requires further exploration in future research.

Proper education about food labels is essential, as labels have the potential to guide consumer behavior towards more sustainable consumption [77]. Food labeling is crucial in promoting public health [79]. Education helps when assessing the nutritional content of a product and in raising awareness of potentially harmful ingredients (allergens and additives) [80,81]. The food industry has to be transparent in their labeling to help facilitate sustainable consumption.

Healthy food habits are shaped not only at school or university but also through mass media and at home. Family and friends play a crucial role in it; children’s eating behaviors are determined by parents [82], and as they grow up, the influence of parents weakens, but friends and colleagues start to matter because meals are usually shared with other people [53]. Healthy eating habits also make healthy food shopping easier. For example, when in stores, the heuristic ‘Buy what I usually do’ can be used [83]. The equalization of socio-cultural and economic influences in Poland might explain the lack of other significant differences in consumer behaviors across generations (in the case of promotions, habit, family/friends’ opinions, packaging appearance, and product advertising). Globalization and easy access to a variety of products could blur traditional generational differences in purchasing preferences. Additionally, the rise in ecological and health awareness across all age groups encourages the search for high-quality products and the consideration of ecological certifications. Moreover, the ubiquitous access to technology and media means that all generations are similarly influenced by marketing and advertising, leading to the homogenization of purchasing behaviors. In such an environment, the diversity of available products allows consumers to make similar purchasing choices, regardless of age.

As we wrote in the introduction, the CBOS study indicated that due to inflation, Poles are buying in advance [58]. Such behavior may stem from the fear that products will soon become more expensive, and that it is worth stocking up; unfortunately, such behavior often leads to food waste as well. Proper education in this area can guard against economic challenges such as inflation, which may trigger less rational consumer behavior. What is interesting in our study is that as many as 82.8% declare that they only buy what they currently need. The oldest generation (Baby Boomers) most frequently adheres to this behavior. This result aligns with earlier studies conducted in our country, indicating that older individuals less often purchase food products they did not plan to buy compared to younger ones [84]. The results of our study are quite interesting because there is a common perception that Baby Boomers engage in excessive consumption compared to younger generations and exhibit little concern for the environment [85]. Our study does not confirm these assertions. Even though the youngest generation grew up in the wake of a climate catastrophe, according to our research, the oldest generation, Baby Boomers, demonstrates a greater inclination toward sustainable customer behaviors. This may stem from the specific upbringing and life circumstances in post-war, socialist, and post-socialist Poland. Baby Boomers had to be frugal and value their possessions for an extended period due to shortages. It is also a generation accustomed to cooking meals at home, while younger generations more frequently dine out [72].

Importantly, our study is not isolated in its conclusions, as other studies have drawn similar conclusions regarding the greater sustainable customer behavior of Baby Boomers Generation. In a study conducted on over 2000 UK consumers, it was indicated that in many cases, the oldest generation is better at eco-activities than Generation Z [86]. The sustainability of the Baby Boomer generation is also emphasized in the study by Severo et al. [87]. In the study conducted by Kamenidou et al. [88], it was found that while organic food is purchased occasionally by all generations, the Silent Generation, the Baby Boomer generation, and Generation X buy organic food more frequently compared to Generation Z and Generation Y.

The primary limitation of the study was the chosen research methodology. The online survey format necessitated the use of a quota sample, as it was not possible to obtain a representative sample given that not all Poles have internet access [89]. Moreover, the most frequently excluded group from using this medium consists primarily of the elderly [89]. Therefore, we must acknowledge that our Baby Boomer population is also a specific one—a subpopulation with access to and skills in using the Internet. This factor could have significance for the results we obtained. Another limitation of our study is the fact that we did not examine actual behaviors but only declarations of behavior. Such declarations must be treated with caution because researchers [90,91] point out the existence of the so-called ‘intention–behavioral’ gap, which is the difference between what consumers declare they will buy and what they actually do. It can appear in socially responsible purchasing declarations as more socially desirable.

Another limitation of our study was the consideration of food as a whole, although we are aware of existing differences in buying behaviors, for example, for dairy, meat, and dry rice or groats. We do not know which of these products the respondents primarily had in mind while answering our questions.

Our article has marketing and policy implications. First of all, it shows what type of rhetoric companies can use in their food advertisements if they want to address their campaigns to different generations. For example, if they want to sell products to Baby Boomers, the domestic origin should be emphasized, while for the youngest generation, Generation Z, focusing on the price will be more effective.

In Poland, there are various groups working towards sustainable food consumption (non-governmental organizations, producers, politicians, and consumers). Developing healthy eating habits among Poles should be actively promoted through comprehensive educational initiatives implemented from the earliest years of school and supported by media campaigns. It is crucial to instill a strong foundation of nutritional knowledge, emphasizing the importance of a balanced diet, understanding food sources, and making informed choices regarding food consumption. Decision makers and public policymakers have to tailor educational campaigns to different generations, as different motives guide them in food choices. They should promote avoiding stockpiling, the consumption of local and seasonal products, packaging-free options, and shortening the distance between the consumer and the producer. Another supported solution should be transparent labeling of food products.

## 6. Conclusions

Different age cohorts of Poles grew up, were raised, and lived in slightly different conditions, shaping their behaviors and lifestyles, including those related to consumption. Our study has shown that there are both similarities and differences in food choices and sustainable consumption behaviors across generations. In line with our hypotheses, we found that, for example, the price of a food product is more important for the youngest generation, but the country of origin is a more important motive of buying food for Baby Boomers. Understanding these kinds of differences can be crucial for creating effective marketing and educational strategies regarding healthy eating and sustainable consumption targeted toward specific generations.

Despite the common belief that the Baby Boomer generation engages in excessive consumption and shows little interest in environmental issues, unlike the youngest generation, which consumes more consciously due to better pro-ecological education, our results suggest otherwise. The initial hypotheses were not confirmed. It turns out that Baby Boomers in Poland exhibit a greater inclination towards sustainable consumer practices, such as checking expiration dates on products and focusing on purchases aligned with current needs. On the other hand, representatives of Generation Z, despite their pro-ecological views, are less likely than might be expected to engage in sustainable consumption.

However, once again, we need to emphasize that we studied a specific group—Internet users—and that this group is not representative of all Poles. Additionally, we may also be dealing with an ‘intention–behavioral’ gap here, as we examined declarations.

In the future, it would be worthwhile to deepen the obtained results, for example, through in-depth interviews, which could provide a more comprehensive understanding of motivation differences, offer more detailed explanations of food choice motives, and shed light on real consumer behaviors. To overcome the limitation of relying on self-reported behaviors, considering alternative techniques such as observation or experimental methods can be taken into account and it is advisable to research products groups separately to avoid interpretative uncertainties.

## Figures and Tables

**Table 1 foods-13-00521-t001:** Motives of buying food by each generation.

When Buying Food, I Am Guided Primarily by…	Z up to 25 Years %	Y 26–42 Years Old %	X 43–57 Years Old %	Baby Boomers 58–76 Years Old %	Total	Chi^2^	*p*
Product quality	64.8	64.8	68.0	78.4	69.0	14.568	**0.002**
Price	73.6	73.6	64.8	62.4	68.6	11.940	**0.008**
Information about the product’s composition provided on the packaging	43.8	40.8	40.8	47.6%	41.0%	8.483	**0.037**
Promotions	31.6	34.8	30.8	25.6	30.7	5.128	0.163
Habit	28.8	24.8	20.4	23.2	24.3	5.018	0.171
Country of origin	12.0	13.6	21.6	23.6	17.7	17.045	**<0.001**
The presence of an ecological certification for the product	7.2	8.8	8.4	14.4	9.7	8.802	**0.032**
Opinions of family/friends	11.2	10.8	8.4	6.8	9.3	3.829	0.281
Packaging appearance	6.0	6.0	6.4	2.2	5.1	6.674	0.083
Product advertising	3.6	2.2	3.2	0.8	2.7	4.682	0.197
Something else	1.2	0.0	0.4	0.4	0.5	3.819	0.282

Bold font indicates statistical significance. N = 1000.

**Table 2 foods-13-00521-t002:** Declarations of sustainable consumer behaviors of individual generations.

	Z up to 25 Years		Y 26–42 Years Old		X 43–57 Years Old		Baby Boomers 58–76 Years Old		Total		Statistic Test	
	% (n) *	M (SD)	% (n) *	M (SD)	% (n) *	M (SD)	% (n) *	M (SD)	% (n) *	M (SD)	F Test	*p*
Before purchasing, I check the expiration date of the product.	80.4 (201)	4.16 (1.02)	85.2 (213)	4.36 (0.81)	85.2 (213)	4.27 (0.85)	92.8 (232)	4.48 (0.678)	85.9 (859)	4.32 (0.88)	5.97	**<0.001**
I buy thoughtfully and only as much as I need at the moment.	77.6 (194)	3.94 (0.90)	82.0 (205)	4.00 (0.88)	83.6 (209)	4.09 (0.824)	88.0 (220)	4.23 (0.718)	82.8 (828)	4.07 (0.84)	5.76	**<0.001**
I am willing to pay 10% more and choose a product labeled as eco-friendly than a cheaper product without such labeling	37.2 (93)	3.02 (1.12)	40.8 (102)	3.12 (1.09)	40.8 (102)	3.08 (1.10)	42.0 (105)	3.16 (1.13)	40.2 (402)	3.09 (1.11)	0.72	0.542
Before purchasing, I check whether the product or its packaging is biodegradable (subject to recycling).	37.6 (94)	2.99 (1.18)	38.8 (97)	2.96 (1.17)	39.6 (99)	3.09 (1.11)	43.2 (108)	3.21 (1.07)	39.8 (398)	3.06 (1.13)	2.43	0.064

* We have added the responses ‘definitely yes’ and ‘rather yes’. Bold font indicates statistical significance. N = 1000.

**Table 3 foods-13-00521-t003:** Post hoc tests for generational groups and declaration of sustainable customer behaviors (MD = mean difference).

	Comparison Group	MD	SE	*p*
Before purchasing, I check the expiration date of the product (Games–Howell post hoc)				
Z up to 25 years	Y 26–42 years old	−0.20	0.09	0.103
	X 43–57 years old	−0.10	0.08	0.605
	Baby Boomers 58–76 years old	−0.32	0.08	**<0.001**
Y 26–42 years old	X 43–57 years old	0.09	0.08	0.640
	Baby Boomers 58–76 years old	−0.12	0.07	0.327
X 43–57 years old	Baby Boomers B 58–76 years old	−0.21	0.07	**0.012**
I buy thoughtfully and only as much as I need at the moment (Tukey post hoc)				
Z up to 25 years	Y 26–42 years old	−0.06	0.07	0.826
	X 43–57 years old	−0.15	0.07	0.174
	Baby Boomers 58–76 years old	−0.29	0.07	**<0.001**
Y 26–42 years old	X 43–57 years old	−0.09	0.07	0.639
	Baby Boomers 58–76 years old	−0.23	0.07	**0.012**
X 43–57 years old	Baby Boomers 58–76 years old	0.14	0.07	0.237

Bold font indicates statistical significance. N = 1000.

**Table 4 foods-13-00521-t004:** Sustainable consumer behavior index of the studied generations.

	Z up to 25 Years		Y 26–42 Years Old		X 43–57 Years Old		Baby Boomers 58–76 Years Old		Total		Statistic Test	
	% (n)	M (SD)	% (n)	M (SD)	% (n)	M (SD)	% (n)	M (SD)	% (n)	M (SD)	F Test	*p*
Index of sustainable consumer behaviors based on the four statements above (scale of 1–5)	25.0 (250)	3.53 (0.70)	25.0 (250)	3.61 (0.69)	25.0 (250)	3.63 (0.61)	25.0 (250)	3.77 (0.61)	100.0 (1000)	3.63 (0.66)	5.87	**<0.001**

Bold font indicates statistical significance. N = 1000.

**Table 5 foods-13-00521-t005:** Games–Howell post hoc post hoc tests for generational groups and **sustainable customer behavior index** (MD = mean difference).

	Comparison Group	MD	SE	*p*
Z up to 25 years	Y 26–42 years old	−0.08	0.06	0.529
	X 43–57 years old	−11	0.06	0.273
	Baby Boomers 58–76 years old	−0.24	0.06	**<0.001**
Y 26–42 years old	X 43–57 years old	−0.02	0.06	0.982
	Baby Boomers 58–76 years old	−0.16	0.06	**0.035**
X 43–57 years old	Baby Boomers B 58–76 years old	−0.14	0.05	0.06

Bold font indicates statistical significance. N = 1000.

## Data Availability

The original contributions presented in the study are included in the article, further inquiries can be directed to the corresponding author.

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
