# Peer review of "Generational Differences in Food Choices and Consumer Behaviors in the Context of Sustainable Development"

_foods, 2024, doi:10.3390/foods13040521_

Round 1

Reviewer 1 Report

Comments and Suggestions for Authors

This paper focuses on generational differences in food choices and consumer behaviors, but there are several critical issues.

1. In the introduction, there is no theoretical contribution. It is directly linked to selling points for readers.

2. Unfortunately, this paper has no research hypotheses at all. Probably, it is related to my early points: theoretical foundation.

3. Becaues this study has no any hypotheses, data anakyses are poor. I have no idea what the authors tell us.

4. Overall, the authors should demonstrate the importance of the paer, its theoretical contribution, and robust analytical results.

Author Response

Dear Reviewer 1,

Thank you for the review of our article ‘Generational Differences in Food Choices and Consumer Behaviors in the Context of Sustainable Development’. Your comments certainly improved our manuscript. With this letter I am submitting a version of the article which has been revised along the lines recommended by you and the others reviewers. My answers to the comments received are in red italics below.

Comment #1

In the introduction, there is no theoretical contribution. It is directly linked to selling points for readers.

The whole new part with theoretical contribution has been added. See lines: 62–90, 97–106.

Comment #2

Unfortunately, this paper has no research hypotheses at all. Probably, it is related to my early points: theoretical foundation

The whole new part with theoretical contribution and hypothesis has been added.

See lines: 126–199, 203–241.

Comment #3 

Because this study has no any hypotheses, data analysis are poor. I have no idea what the authors tell us.

In accordance with the reviewer's suggestion, the article was supplemented explicitly with the hypotheses initially accepted implicitly. Their verification is presented by the analyses included in the article. See lines: 126–199, 203–241, 244–254, 259–265, 278, 282, 338–342, 346–349, 361–380, 410–421.

Comment #4

Overall, the authors should demonstrate the importance of the paper, its theoretical contribution, and robust analytical results.

All suggestions of the reviewer have been taken into account. The importance of the article has been highlighted, we have added theoretical framework and hypotheses to which the results of the analysis are subordinated. See lines: 126–199, 203–241, 244–254, 259–265, 278, 282, 338–342, 346–349, 361–380, 410–421.

I hope that these changes are acceptable. If you have any further comments we would be happy to address them.

Thank you!

Reviewer 2 Report

Comments and Suggestions for Authors

1.The Introduction section of this study has several deficiencies and areas that could be improved.

(1)The introduction presents sustainable development in the context of food but does so rather briefly. Expanding on how sustainable food systems contribute to social equity would offer a more nuanced understanding of the subject.

(2)The exploration into generational cohorts introduces the concept effectively but lacks an in-depth analysis of historical and social contexts that shape these cohorts’ consumption behaviors.

(3)The positioning of the study within Polish society is a strength, yet the introduction could further highlight how Poland's unique socio-political backdrop influences consumer behavior, providing a richer context for the study's focus.

2.The Materials and Methods section of this study has several deficiencies and areas that could be improved.

(1)The questionnaire, a critical tool for data collection, is described as proprietary and developed by the authors, yet the section leaves a gap in understanding by not discussing the theoretical framework or the validation process of the survey questions. The robustness and repeatability of results are linked to the validity and reliability of the instrument used. Without this information, the quality of the data gathered through the questionnaire could be questioned.

(2)Although a pilot study was conducted, the sample size of 10 individuals may not be sufficient to fully vet and validate the questionnaire. Increasing the pilot study's sample size could improve the credibility of the instrument.

3.The Results section of this study has several deficiencies and areas that could be improved.

(1)The lack of differentiation in certain behaviors based on generational affiliation is mentioned but not discussed in depth. It would be instructive to explore why some behaviors do not vary significantly by age group, as this could have implications for understanding consumer behavior patterns.

(2)There may be some inconsistency in the terms used to reference generational cohorts (e.g., labeling 'BB' for 'Baby Boomers') which could potentially confuse readers; consistency in labels across all sections is paramount to avoid misinterpretation.

4.The Discussion section of this study has several deficiencies and areas that could be improved.

(1)While the study indicates clear generational differences in motivations for buying food, the discussion seems to generalize consumer behavior without offering a deeper analysis of the underlying reasons behind these purchasing habits, which could lead to a simplistic interpretation of complex behaviors.

(2)When discussing the importance of checking expiration dates, the exploration of actual understanding and consumer education regarding food safety could be expanded to provide a more complete picture of the behavior.

(3)The study connects the declaration of not buying in excess to inflationary responses. Additional discussion on consumer education regarding sustainable consumption could offer a balanced view of how behavior changes amid economic stresses.

(4)While the study challenges stereotypes, further discussion could account for possible disparities between declared behaviors and actual purchasing patterns, as this discrepancy is notable in consumer behavior research.

5.The Conclusions section of this study has several deficiencies and areas that could be improved.

(1)The conclusion that Baby Boomers in Poland exhibit a greater inclination towards sustainable consumer practices might be too broad. The generational behaviors discussed in the findings are specific to the study's sample and may not be universally representative of all Poles within these age cohorts.

(2)The role of social desirability bias in self-reported data is not discussed. Since this can affect how participants report their behaviors, particularly in the domain of sustainable practices, an acknowledgment of this potential bias would fortify the conclusions drawn.

Author Response

Dear Reviewer 2,

Thank you for the review of our article ‘Generational Differences in Food Choices and Consumer Behaviors in the Context of Sustainable Development’. Your comments certainly improved our manuscript. With this letter I am submitting a version of the article which has been revised along the lines recommended by you and the others reviewers. My answers to the comments received are in red italics below.

Introduction Comment #1

The introduction presents sustainable development in the context of food but does so rather briefly. Expanding on how sustainable food systems contribute to social equity would offer a more nuanced understanding of the subject.

Following the reviewer's advice, in the introduction we developed theoretical considerations on how sustainable food systems contribute to social equity. See lines: 62–90.

Introduction Comment #2

The exploration into generational cohorts introduces the concept effectively but lacks an in-depth analysis of historical and social contexts that shape these cohorts’ consumption behaviors.

The historical and social context that shaped the consumption behavior of each generation was added. See lines: 203–219.

Introduction Comment #3

The positioning of the study within Polish society is a strength, yet the introduction could further highlight how Poland's unique socio-political backdrop influences consumer behavior, providing a richer context for the study's focus.

Information on the specifics of Poland and the socio-political context, as a post-communist country, was developed in accordance with the reviewer's comment. See lines: 203–219.

The Materials and Methods Comment #1

The questionnaire, a critical tool for data collection, is described as proprietary and developed by the authors, yet the section leaves a gap in understanding by not discussing the theoretical framework or the validation process of the survey questions. The robustness and repeatability of results are linked to the validity and reliability of the instrument used. Without this information, the quality of the data gathered through the questionnaire could be questioned.

We have developed the Materials and Methods section and included comments about validation of the questionnaire. While creating the questionnaire we followed the procedures outlined by Malhotra (1999), and we added this information to the main text. See lines: 244–254.

 The Materials and Methods Comment #2

Although a pilot study was conducted, the sample size of 10 individuals may not be sufficient to fully vet and validate the questionnaire. Increasing the pilot study's sample size could improve the credibility of the instrument.

Thank you for this comment, we have changed the word pilot study to pretesting, in line with the distinction made by Smith and Albaum (2005). The issue of questionnaire validation has already been addressed before. See lines: 244–254.

The Results Comment #1

The lack of differentiation in certain behaviors based on generational affiliation is mentioned but not discussed in depth. It would be instructive to explore why some behaviors do not vary significantly by age group, as this could have implications for understanding consumer behavior patterns.

Thank you for this remark, it is indeed an interesting issue. We have decided to develop this thread in the Discussion section, see lines: 422–450.

The Results Comment #2

There may be some inconsistency in the terms used to reference generational cohorts (e.g., labeling 'BB' for 'Baby Boomers') which could potentially confuse readers; consistency in labels across all sections is paramount to avoid misinterpretation.

We have corrected this imperfection in the text. We now use the term Baby Boomers everywhere.

The Discussion Comment #1

While the study indicates clear generational differences in motivations for buying food, the discussion seems to generalize consumer behavior without offering a deeper analysis of the underlying reasons behind these purchasing habits, which could lead to a simplistic interpretation of complex behaviors.

The reasons behind purchasing habits, were given further thought. See lines:427–450.

The Discussion Comment #2

When discussing the importance of checking expiration dates, the exploration of actual understanding and consumer education regarding food safety could be expanded to provide a more complete picture of the behavior.

The need for consumer education regarding food safety was discussed, in the passage of the text indicated by the reviewer and elsewhere. See lines: 491–502.

The Discussion Comment #3

The study connects the declaration of not buying in excess to inflationary responses. Additional discussion on consumer education regarding sustainable consumption could offer a balanced view of how behavior changes amid economic stresses.

In the discussion, following the reviewer's advice, we have also included an element of consumer education on sustainable consumption and how this can influence how people's behavior changes when economic challenges such as inflation occur. See lines: 427–461.

The Discussion Comment #4

While the study challenges stereotypes, further discussion could account for possible disparities between declared behaviors and actual purchasing patterns, as this discrepancy is notable in consumer behavior research.

Thank you for this accurate remark. In the discussion, we elaborated on the differences between behavioral declarations and actual behavior. See lines: 476–481.

The Conclusions Comment #1

The conclusion that Baby Boomers in Poland exhibit a greater inclination towards sustainable consumer practices might be too broad. The generational behaviors discussed in the findings are specific to the study's sample and may not be universally representative of all Poles within these age cohorts.

Thank you for this remark. We rephrase the conclusion after taking Reviewer observation into account. See lines:506–523, 527–530.

The Conclusions Comment #2

 The role of social desirability bias in self-reported data is not discussed. Since this can affect how participants report their behaviors, particularly in the domain of sustainable practices, an acknowledgment of this potential bias would fortify the conclusions drawn.

The role of social desirability bias in self-reported data has been added to conclusion section according to Reviewer suggestion. See lines: 519–530.

I hope that these changes are acceptable. If you have any further comments we would be happy to address them.

Thank you!

Reviewer 3 Report

Comments and Suggestions for Authors

The manuscript investigates Polish consumers' food-related preferences based on generational groups.

Though the topic per sé has importance, there are some major shortcomings that should be addressed.

1) The rationale of ex-ante clustering the respondents is not explained. Why not simply clustering based on age, and - if relevant - then check whether clusters are in line with the different generation age categories.

2) In addition to the previous point, Table 1 provides no clear information on which values are statistically significantly proven to be different between the respective generations. Without this, forcing it to be clustered based on generational differences makes no sense.

3) I heavily miss validation of why and how the attributes and statements investigated were selected.

4) My biggest concern is that the survey tries to collect data on 'food', in general. Obviously, there are substantial differences between different product groups (e.g., dairy, fresh meat vs. dried pasta) in terms of consumer attitudes; therefore, results can not really be interpreted.

5) I miss dedicated sections for managerial and policy implications.

Author Response

Dear Reviewer 3,

Thank you for the review of our article ‘Generational Differences in Food Choices and Consumer Behaviors in the Context of Sustainable Development’. Your comments certainly improved our manuscript. With this letter I am submitting a version of the article which has been revised along the lines recommended by you and the others reviewers. My answers to the comments received are in italics below.

Comment #1

The rationale of ex-ante clustering the respondents is not explained. Why not simply clustering based on age, and - if relevant - then check whether clusters are in line with the different generation age categories.

The reasons of ex-ante clustering have been described. See lines: 259–265.

Comment #2

In addition to the previous point, Table 1 provides no clear information on which values are statistically significantly proven to be different between the respective generations. Without this, forcing it to be clustered based on generational differences makes no sense.

Following the reviewer's advice, in Table 1 we have clearly indicated (bold) where there are differences between the generations.   See lines: 314–315.

Comment #3

I heavily miss validation of why and how the attributes and statements investigated were selected.

Following the advice of the reviewer, information about the validation of the questionnaire have been included. Throughout the development of the questionnaire we followed the procedures outlined by Malhotra (1999) and this information has been seamlessly integrated into the main text. See lines: 247–254.

Comment #4

My biggest concern is that the survey tries to collect data on 'food', in general. Obviously, there are substantial differences between different product groups (e.g., dairy, fresh meat vs. dried pasta) in terms of consumer attitudes; therefore, results can not really be interpreted.

Thank you for this accurate comment. What the reviewer pointed out has been added to the limitations of the research section. See lines: 482–485.

Although, we believe that the interpretation of our results is reasonable, as even dried food in Poland now has expiration dates and can be purchased in ecological packaging (or without any packaging at all). They may be more expensive if they are organic, but it is, of course, more prudent to buy them in advance, as the expiration dates are longer than on dairy products.

Comment #4

I miss dedicated sections for managerial and policy implications

According to Reviewer suggestion, sections for managerial and policy implications has been added. See lines: 486–502.

I hope that these changes are acceptable. If you have any further comments we would be happy to address them.

Thank you!

Round 2

Reviewer 1 Report

Comments and Suggestions for Authors

This revision is fine. Well done.

Reviewer 2 Report

Comments and Suggestions for Authors

The text has undergone extensive revisions following feedback from the reviewer. The authors have addressed my comments and suggestions, significantly improving the quality and clarity of this manuscript. Consequently, I am prepared to recommend an acceptance.

Reviewer 3 Report

Comments and Suggestions for Authors

My concerns were addressed.